# Transfer Learning of Graph Neural Networks with Ego-graph Information Maximization

## Abstract

Graph neural networks (GNNs) have been shown with superior performance in various applications, but training dedicated GNNs can be costly for large-scale graphs. Some recent work started to study the pre-training of GNNs. However, none of them provide theoretical insights into the design of their frameworks, or clear requirements and guarantees towards the transferability of GNNs. In this work, we establish a theoretically grounded and practically useful framework for the transfer learning of GNNs. Firstly, we propose a novel view towards the *essential graph information* and advocate the capturing of it as the goal of transferable GNN training, which motivates the design of EGI (*ego-graph information maximization*) to analytically achieve this goal. Secondly, we specify the requirement of *structure-respecting* node features as the GNN input, and conduct a *rigorous analysis of GNN transferability* based on the difference between the local graph Laplacians of the source and target graphs. Finally, we conduct controlled synthetic experiments to directly justify our theoretical conclusions. Extensive experiments on real-world networks towards role identification show consistent results in the rigorously analyzed setting of direct-transfering (freezing parameters), while those towards large-scale relation prediction show promising results in the more generalized and practical setting of transfering with fine-tuning.

## 1 Introduction

Graph neural networks (GNNs) have been intensively studied recently (Kipf & Welling, 2017; Keriven & Peyré, 2019; Chen et al., 2019; Oono & Suzuki, 2020; Huang et al., 2018), due to their established performance towards various real-world tasks (Hamilton et al., 2017; Ying et al., 2018b; Velickovic et al., 2018), as well as close connections to spectral graph theory (Defferrard et al., 2016; Bruna et al., 2014; Hammond et al., 2011). While most GNN architectures are not very complicated, the training of GNNs can still be costly regarding both memory and computation resources on real-world large-scale graphs (Chen et al., 2018; Ying et al., 2018a). Moreover, it is intriguing to transfer learned structural information across different graphs and even domains in settings like few-shot learning (Vinyals et al., 2016; Finn et al., 2017; Ravi & Larochelle, 2017). Therefore, several very recent studies have been conducted on the transferability of GNNs, which focus on the setting of pre-training plus fine-tuning (Hu et al., 2019a,b, 2020; Wu et al., 2020). However, it is unclear in what situations the models will excel or fail especially when the pre-training and fine-tuning tasks are different. To provide rigorous analysis and guarantee on the transferability of GNNs, we focus on the setting of direct-transfering between the source and target graphs, under an analogous setting of "domain adaptation" (Ben-David et al., 2007).

In this work, we establish a theoretically grounded framework for the transfer learning of GNNs, and leverage it to design a practically transferable GNN model. Figure 1 gives an overview of our framework. It is based on a novel view of a graph as samples from the joint distribution of its k-hop ego-graph structures and node features, which allows us to define graph information and similarity, so as to analyze GNN transferability (§2). This view motivates us to design EGI, a novel GNN model based on ego-graph information maximization, which is effective in capturing the graph information as we define (§2.1). Then we further specify the requirement on transferable node features and analyze the transferability of EGI that is dependent on the local graph Laplacians of source and target graphs (§2.2).

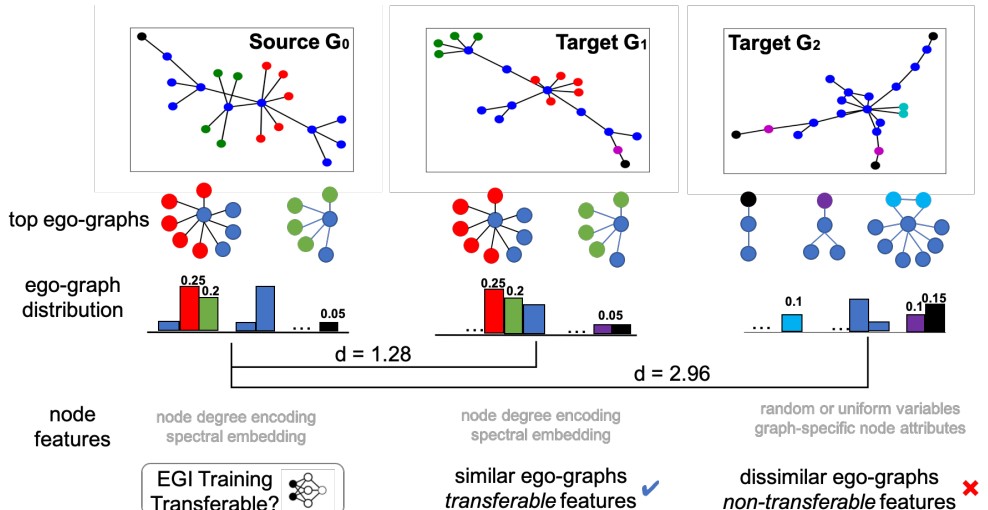

**Figure 1:** Overview of our GNN transfer learning framework: (1) we represent graph as a combination of its 1-hop ego-graph and node feature distributions; (2) we design a transferable GNN regarding the capturing of such essential graph information; (3) we establish a rigorous guarantee of GNN transferability based on the requirement on nodes features and difference between graph structures.

All of our theoretical conclusions have been directly validated through controlled synthetic experiments (Table 1), where we use structural-equivalent role identification in a direct-transfering setting to analyze the impacts of different model designs, node features and source-target structure similarities on GNN transferability. In §3, we conduct real-world experiments on multiple publicly available network datasets. On the Airport and Gene graphs (§3.1), we closely follow the settings of our synthetic experiments and observe consistent but more detailed results supporting the design of EGI and the utility of our theoretical analysis. On the YAGO graphs (§3.2), we further evaluate EGI on the more generalized and practical setting of transfer learning with task-specific fine-tuning. We find our theoretical insights still indicative in such scenarios, where EGI consistently outperforms state-of-the-art GNN models and transfer learning frameworks with significant margins.

## 2 TRANSFERABLE GRAPH NEURAL NETWORKS

Based on the connection between GNN and spectral graph theory (Kipf & Welling, 2017), we describe the output of a GNN as a combination of its input node features, fixed graph Laplacian and learnable graph filters. The goal of training a GNN is then to improve its utility by learning the graph filters that are compatible with the other two components towards specific tasks.

In the graph transfer learning setting where downstream tasks are often unknown during pre-training, we argue that the general utility of a GNN should be optimized and quantified *w.r.t.* its ability of capturing the essential graph information in terms of the joint distribution of its link structures and node features, which motivates us to design a novel ego-graph information maximization model (EGI) (§2.1). The general transferability of a GNN is then quantified by the gap between its abilities to model the source and target graphs. Under reasonable requirements such as using *structure-respecting* node features as the GNN input, we analyze this gap for EGI based on the structural difference between two graphs *w.r.t.* their local graph Laplacians (§2.2).

### 2.1 TRANSFERABLE GNN VIA EGO-GRAPH INFORMATION MAXIMIZATION

In this work, we focus on the *direct-transfering setting* where a GNN is pre-trained on a source graph $G_a$ in an unsupervised fashion and applied on a target graph $G_b$ without fine-tuning.[1] Consider a graph $G = \{V, E\}$, where the set of nodes $V$ are associated with certain features and the set of links $E$ form certain structures. Intuitively, the transfer learning will be successful only if both the features and structures of $G_a$ and $G_b$ are similar in some ways, so that the graph filters of a GNN learned on $G_a$ are compatible with the features and structures of $G_b$.

---

[1]In the experiments, we show our model to be generalizable to the more practical settings with task-specific pre-training and fine-tuning, while the study of rigorous bound in such scenarios is left as future work.

Motivated by the concept of k-layer expansion sub-graph in (Bai & Hancock, 2016), we introduce a novel view of a graph as *samples from the joint distribution of its k-hop ego-graph structures and node features*. This view allows us to give concrete definitions towards *structural information* of graphs in the transfer learning setting, which facilitates the measuring of similarity (difference) among graphs.

**Definition 2.1** (K-hop ego-graph). *We call a graph $g_i = \{V(g_i), E(g_i)\}$ a k-hop ego-graph centered at $v_i$ if it has a k-layer centroid expansion (Bai & Hancock, 2016) such that the greatest shortest path rooted from $v_i$ has length k, i.e., $k = \max_{v_j \in V} |S(v_i, v_j)|$, where $S(v_i, v_j)$ is the shortest path between $v_i$ and $v_j$.*

For an ordered k-hop ego-graph, we denote $v_{p,q}$ as the $q$-th node in the $p$-th layer of the ego-graph (*i.e.*, $|S_i(v_i, v_{p,q})| = p$), where $p = 0, \ldots, k$, and $e_{vv'}$ as the edge between $v_{p,q}$ and $v_{p+1,q'}$.

**Definition 2.2** (Structural information). *Let $\mathcal{G}$ be a topological space of sub-graphs (Verma & Zhang, 2019). We view a graph $G$ as samples of k-hop ego-graphs $G = \{g_i\}_{i=1}^n$ drawn i.i.d. from $\mathcal{G}$ with probability $\mu$, i.e., $g_i \overset{\text{i.i.d.}}{\sim} \mu \ \forall i = 1, \cdots, n$. The structural information of $G$ is then defined to be the combination of the distribution $\mu$ and the set of spectrum of $\{g_i\}_{i=1}^n$.*

The structural information of a graph $G$ can be characterized by $\{g_i\}_{v_i \in V}$ and its empirical distribution, where each $g_i$ is a k-hop ego-graph of $G$ centered at node $v_i$ with $V(g_i) = \{u \in V(G) : S(u, v_i) \le k\}$, and edges $E(g_i) = \{e_{uv} \in E(G) : u, v \in V(g_i)\}$. As shown in Figure 1, three graphs $G_0$, $G_1$ and $G_2$ are characterized by a set of 1-hop ego-graphs and their empirical distributions, which allows us to quantify the structural similarity among graphs as shown in §2.2 (*i.e.*, $G_0$ is more similar to $G_1$ than $G_2$ under such characterization).

In practice, the nodes in a graph $G$ are characterized not only by their k-hop ego-graph structures but also their associated node features. Therefore, $G$ should be regarded as samples $\{(g_i, x_i)\}^n \in \mathcal{G} \times \mathcal{X}$, drawn with the joint distribution $p$ on the product space of $\mathcal{G}$ and a node feature space $\mathcal{X}$. To capture such joint distributions of structural information and node features, we design *ego-graph information maximization* (EGI), which recursively reconstructs the k-hop ego-graph of each node based on their features in an unsupervised fashion.

**Ego-Graph Information Maximization.** Assume we are given a set of ego-graphs $\{(g_i, x_i)\}_i$ with empirical joint distribution $\mathbb{P}$. Similarly with the "local" version of DIM (Hjelm et al., 2019), we define $\mathbb{U}_{\Psi(g_i, x_i)}$ as the empirical distribution of the embedding produced by the GNN encoder $\Psi$ for the the center node $v_i$ of ego-graph $g_i$. Unlike DGI (Velickovic et al., 2019) that models the local-global mutual information (MI), EGI optimizes $\Psi$ to maximize the MI of $\mathcal{I}(g_i, \Psi(g_i, x_i))$, which is directly between the structural input and output of GNN, with a focus on the structural information $g_i$. Specifically, we use the Jensen-Shannon MI estimator in (Hjelm et al., 2019),

$$\mathcal{L}_{\text{EGI}} = -\mathcal{I}^{\text{(JSD)}}(\mathcal{G}, \Psi) = \mathbb{E}_{\mathbb{P} \times \tilde{\mathbb{U}}}\left[\text{sp}\left(T_{\mathcal{D}, \Psi}(g_i, \Psi(g_i', x_i'))\right)\right] - \mathbb{E}_{\mathbb{P}}\left[-\text{sp}\left(-T_{\mathcal{D}, \Psi}(g_i, \Psi(g_i, x_i))\right)\right], \quad (1)$$

$T_{\mathcal{D}, \Psi} = \mathcal{D} \circ (g_i, \Psi(g_i, x_i))$, where $\mathcal{D}$ is a discriminator $\mathcal{D} : g_i \times \Psi(g_i, x_i) \to \mathbb{R}^+$. In Eq. 1, during the training of $\mathcal{D}$, the input space of $\mathcal{D}$ is at least as large as the number of graph permutations $|V(g_i)|!$. Instead of enumerating all possible graphs $g_i'$, we fix $g_i$ and sample GNN's output $\Psi(g_i', x_i')$ from the marginal distribution $\tilde{\mathbb{U}}$ by uniformly sampling $(g_i', x_i') \sim \tilde{\mathbb{P}}, \tilde{\mathbb{P}} = \mathbb{P}$. The correspondence between sampling $(g_i', x_i') \sim \tilde{\mathbb{P}}$ and $g_i' \sim \mathcal{G}$ is discussed in Remark 2 when node features are strcuture-respecting (Def. 2.3).

Formally, we characterize the decision process of $\mathcal{D}$ with a fixed graph ordering, *i.e.*, BFS-ordering $\pi$ over edges $E(g_i)$. $\mathcal{D}$ is a GNN scoring function over an edge sequence $E^\pi : \{e_1, e_2, ..., e_n\}$, which makes predictions on BFS-ordered edges. Let $z_i = \Psi(g_i, x_i)$ , then we have,

$$\mathcal{D}(g_i, z_i) = \sum_{p=0}^{k} \sum_{q=1}^{|V_p(g_i)|} \log \mathcal{D}(e_{\tilde{v}v} | h_{p,q}^{\tilde{q}}, x_{p,q}^i, z_i), \quad (2)$$

where $h$ is the hidden representation output by $\mathcal{D}$, $e_{\tilde{v}v} \in E(g_i)$ is an edge between node $\tilde{v}$ in layer $p$ and $v$ in layer $p + 1$, following the notation defined below Def 2.1. More specifically, we have

$$\mathcal{D}(e_{\tilde{v}v} | h_{p,q}^{\tilde{q}}, x_{p,q}^i, z_i) = \sigma\left(U^T \cdot \tau\left(W^T[h_{p,q}^{\tilde{q}} || x_{p,q}^i || z_i]\right)\right), \quad (3)$$

where $\sigma$ and $\tau$ are Sigmoid and ReLU activation functions, respectively. Thus, the discriminator is asked to distinguish positive $(e_{\tilde{v}v}, \Psi(g_i, x_i))$ and negative pair $(e_{\tilde{v}v}, \Psi(g_i', x_i'))$ that consists of an observed edge and positive/negative center node embeddings $\Psi(\cdot)$.

Due to the fact that the output of a k-layer GNN only depends on a k-hop ego-graphs, EGI can be trained in parallel by sampling batches of $g_i$'s. Besides, the training objective of EGI is transferable as long as $(g_i, x_i)$ across source graph $G_a$ and $G_b$ satisfies the conditions given in §2.2. More details about the model are in Appendix §B and source code in the Supplementary Materials.

**Connection with existing work.** To provide more insights into the EGI objective, we also present it as a dual problem of ego-graph reconstruction. Recall our definition of ego-graph mutual information $\mathcal{I}(g_i, \Psi(g_i, x_i))$. It can be related to an ego-graph reconstruction loss $R(g_i|\Psi(g_i, x_i))$ as

$$\max \mathcal{I}(g_i, \Psi(g_i, x_i)) = H(g_i) - H(g_i|\Psi(g_i, x_i)) \leq H(g_i) - R(g_i|\Psi(g_i, x_i)). \tag{4}$$

When EGI is maximizing the mutual information, it simultaneously minimizes the upper error bound of reconstructing an ego-graph $g_i$. In this view, the key difference between EGI and GVAE (Kipf & Welling, 2016) is they assume each edge in a graph to be observed independently during the reconstruction, while we assume the edges in an ego-graph to be observed jointly. Moreover, existing mutual information based GNNs such as DGI (Velickovic et al., 2019) and GMI (Peng et al., 2020) explicitly measure the mutual information between node features $x$ and GNN output $\Psi$. In this way, they tend to capture node features instead of graph structures, which we deem more essential in graph transfer learning as discussed in §2.2.

**Supportive observations.** In the first three columns of Table 1, in both cases of transfering GNNs between similar graphs (F-F) and dissimilar graphs (B-F), EGI significantly outperforms all competitors when using node degree one-hot encoding as transferable node features. In particular, the performance gains over the untrained GIN and GCN show the effectiveness of training and transfering, and our gains are always larger than the two state-of-the-art unsupervised GNNs. Such results clearly indicate advantageous structure preserving capability and transferability of EGI.

## 2.2 TRANSFERABILITY ANALYSI BASED ON LOCAL GRAPH LAPLACIANS

We now study the transferability of a GNN (in particular, EGI) between the source graph $G_a$ and target graph $G_b$ based on the graph similarity between $G_a$ and $G_b$. We firstly establish the requirement towards node features, under which we then focus on analyzing the transferability of EGI *w.r.t.* the structural information of $G_a$ and $G_b$.

Recall our view of the GNN output as a combination of its input node features, fixed graph Laplacian and learnable graph filters. The utility of a GNN is determined by the compatibility among the three. In order to fulfill such compatibility, we require the node features to be *structure-respecting*:

**Definition 2.3** (Structure-respecting node features). *Let $g_i$ be an ordered ego-graph centered on node $v_i$ with a set of node features $\{x_{p,q}^i\}_{p=0,q=1}^{k,|V_p(g_i)|}$, where $V_p(g_i)$ is the set of nodes in p-th hop of $g_i$. Then we say the node features on $g_i$ are structure-respecting if $x_{p,q}^i = [f(g_i)]_{p,q} \in \mathbb{R}^d$ for any node $v_q \in V_p(g_i)$, where $f : \mathcal{G} \to \mathbb{R}^{d \times |V(g_i)|}$ is a function. In the strict case, $f$ should be injective.*

In its essence, Def 2.3 requires the node features to be a function of the graph structures, which is sensitive to changes in the graph structures, and in an ideal case, injective to the graph structures. In this way, when the learned graph filters of a transfered GNN is compatible to the structure of $G$, they are also compatible to the node features of $G$. As we will explain in Remark 2 of Theorem 2.1, this requirement is also essential for the analysis of our GNN transferability which eventually only depends on the structural difference between two graphs.

In practice, commonly used node features like node degrees, PageRank scores (Page et al., 1999), spectral embeddings (Chung & Graham, 1997), and many pre-computed unsupervised network embeddings (Perozzi et al., 2014; Tang et al., 2015; Grover & Leskovec, 2016) are all structure-respecting in nature. However, other commonly used node features like random vectors (Yang et al., 2019) or uniform vectors (Xu et al., 2019) are not and thus non-transferable. When organic node attributes are available, they are transferable as long as the concept of *homophily* (McPherson et al., 2001) applies, which also implies Def 2.3, but we do not have a rigorous analysis on it yet.

**Supportive observations.** In the fifth and sixth columns in Table 1, where we use uniform embedding as non-transferable node features to contrast with the first three columns, there is almost no or even negative transferability for all compared methods when non-transferable features are used, as the performance of trained GNNs are similar to or worse than their untrained baselines.

With our view of graphs and requirement on node features both established, now we derive the following theorem by characterizing the performance difference of EGI on two graphs based on Eq. 1.

**Theorem 2.1** (GNN transferability). *Let $G_a = \{(g_i, x_i)\}_{i=1}^n$ and $G_b = \{(g_{i'}, x_{i'})\}_{i'=1}^m$ be two graphs. Then denote $L_{g_i}$ as the (normalised) graph Laplacian of $g_i$ $\forall i = 1, \cdots, n$, and let the node features of $g_i$ be structure-respecting and normalized (similarly for $g_{i'}$). Consider GNN $\Psi_\theta$ with $k$ layers and a 1-hop polynomial filter $\phi_\theta$. With reasonable assumptions on the local spectrum of $G_a$ and $G_b$, the empirical performance difference of $\Psi_\theta$ with $\phi_\theta$ evaluated on $\mathcal{L}_{\mathrm{EGI}}$ satisfies*

$$|\mathcal{L}_{\mathrm{EGI}}(G_a) - \mathcal{L}_{\mathrm{EGI}}(G_b)| \leq \mathcal{O}\left(M + \frac{1}{nm}\sum_{i=1}^n\sum_{i'=1}^m \|\lambda(L_{g_i}) - \lambda(L_{g_{i'}})\|_2\right), \qquad (5)$$

*where $M$ is a constant dependant on $k$, $\phi_\theta$, $\{L_{g_i}\}$, $\{L_{g_{i'}}\}$, $\{x_i\}$, $\{x_{i'}\}$, and finally $\lambda(L_{g_i})$ denotes the ordered eigenvalues of the graph Laplacian of $g_i \in G_a$ (similarly for $g_{i'}$).*

*Proof.* The full proof is detailed in Appendix §A. □

**Remark 1.** *Our view of a graph $G$ as samples of $k$-hop ego-graphs is important, as it allows us to make node-wise characterization of GNN similarly as in (Verma & Zhang, 2019). It also allows us to set the depth of ego-graphs in the analysis to be the same as the number of GNN layers $(k)$, since the GNN embedding of each node mostly depends on its $k$-hop ego-graph instead of the whole graph.*

**Remark 2.** *For Eq. 1, Def 2.3 ensures the sampling of GNN embedding at a node always corresponds to sampling an ego-graph from $\mathcal{G}$, which reduces to uniformly sampling from $G = \{g_i\}_{i=1}^n$ under the setting of Theorem 2.1. Therefore, the requirement of Def 2.3 in the context of Theorem 2.1 guarantees the analysis to be only depending on the structural information of the graph.*

The analysis in Theorem 2.1 naturally instantiates our insight about the correspondence between structural similarity and GNN transferability. It tells us how well a GNN trained on $G_a$ can work on $G_b$ by only checking the local graph Laplacians of $G_a$ and $G_b$ without actually training the model.

In practice, the computation of eigenvalues on the small ego-graphs can be rather efficient (Arora et al., 2005), and we do not need to enumerate all pairs of ego-graphs. Suppose we need to sample $M$ pairs of $k$-hop ego-graphs to compare two large graphs, and the average size of ego-graphs are $L$, then the overall complexity of computing Eq. 5 is $\mathcal{O}(ML^2)$, where $M$ is often less than 1K and $L$ less than 50.

**Supportive observations.** In Table 1, in the $\bar{d}$ columns, we compute the average structural difference between two Forest-fire graphs ($\bar{d}(F, F)$) and between Barabasi and Forest-fire graphs ($\bar{d}(B, F)$), based on the RHS of Eq. 5. The results validate our usage of the two graph models to generate structurally different graphs, while also verify our novel view of graphs and the way we propose based on it to characterize structural information of graphs. We further highlight in the $\Delta$ columns the performance difference between the GNNs transferred from Forest-fire graphs and Barabasi graphs to Forest-fire graphs. Since Forest-fire graphs are more similar to Forest-fire graphs than Barabasi graphs (as verified in the $\bar{d}$ columns), we expect $\Delta$ to be positive and large, indicating more positive transfer between the more similar graphs. Indeed, the behaviors of EGI align well with the expectation, which indicates its well-understood transferability and the utility of our theoretical analysis.

**Table 1:** Synthetic experiments of identifying structural equivalent nodes. We randomly generate 40 graphs with the Forest-fire model (F) (Leskovec et al., 2005) and 40 graphs with the Barabasi model (B) (Albert & Barabási, 2002), The GNN models we use include the untrained encoders of GCN (Kipf & Welling, 2017) and GIN (Xu et al., 2019) with random parameters (baselines with only the neighborhood aggregation function), GVAE with GCN encoder (Kipf & Welling, 2016), DGI with GIN encoder (Velickovic et al., 2019), and EGI with GIN encoder. We train GVAE, DGI and EGI on one graph from either set (F and B), and test them on the rest of Forest-fire graphs (F). More details about the results and dataset can be found in Appendix §C.1.

| Method | transferable features | | | non-transferable feature | | | structural difference | |
|---|---|---|---|---|---|---|---|---|
| | F-F | B-F | $\Delta$ | F-F | B-F | $\Delta$ | $\bar{d}$(F,F) | $\bar{d}$(B,F) |
| GCN (untrained) | 0.478 | 0.478 | / | 0.229 | 0.229 | / | | |
| GIN (untrained) | 0.572 | 0.572 | / | 0.358 | 0.358 | / | | |
| GVAE (GCN) | 0.498 | 0.432 | +0.066 | 0.240 | 0.239 | 0.001 | 1.78 | 2.17 |
| DGI (GIN) | 0.578 | 0.591 | -0.013 | 0.394 | 0.213 | +0.181 | | |
| EGI (GIN) | **0.710** | 0.616 | +0.094 | 0.376 | 0.346 | +0.03 | | |

## 3 REAL DATA EXPERIMENTS

**Baselines.** We compare the proposed model with existing unsupervised GNNs and pre-training GNN frameworks. The unsupervised GNNs are the same as used in our synthetic experiments, *i.e.*, GVAE with GCN encoder (Kipf & Welling, 2016) and DGI with GIN encoder (Velickovic et al., 2019). The pre-training GNN frameworks include Mask-GIN and ContextPred-GIN, two node-level pre-training models proposed in (Hu et al., 2019a)[2]. Besides, Structural Pre-train (Hu et al., 2019b) also conducts unsupervised node-level pre-training with structural features like node degrees and clustering coefficients.

**Protocols.** By default, we use node degree one-hot encoding as the transferable feature across all different graphs. As stated before, other transferable features like spectral and other pre-computed node embeddings are also applicable. We focus on the setting where the downstream tasks on target graphs are unspecified but assumed to be structure-relevant, and thus pre-train the GNNs on source graphs in an unsupervised fashion.[3] In terms of evaluation, we design two realistic experimental settings: (1) Direct-transfering on the more structure-relevant task of role identification without given node features to directly evaluate the utility and transferability of EGI. (2) Few-shot learning on relation prediction with task-specific node features to evaluate the generalization ability of EGI.

### 3.1 DIRECT-TRANSFERING ON ROLE IDENTIFICATION

First, we use the role identification without node features in a *direct-transfering* setting as a reliable proxy to evaluate transfer learning performance regarding different pre-training objectives. Role in a network is defined as nodes with similar structural behaviors, such as *clique members*, *hub* and *bridge* (Henderson et al., 2012). Across graphs in the same domain, we assume the definition of role to be consistent, and the task of role identification is highly structure-relevant, which can directly reflect the transferability of different methods and allows us to conduct the analysis according to Theorem 2.1. Upon convergence of pre-training each model on the source graphs, we directly apply them on the target graphs and further train a multi-layer perceptron (MLP) upon their outputs. The GNN parameters are freezing during the MLP training. We refer to this strategy as *direct-transfering* since there is no fine-tuning of the models after transfering to the target graphs.

We use two real-world network datasets with role-based node labels: (1) Airport (Ribeiro et al., 2017) contains three networks from different regions– Brazil, USA and Europe. Each node is an airport and each link is the flight between airports. The airports are assigned with external labels based on their *level of popularity*. (2) Gene (Yang et al., 2019) contains the gene interactions regarding 50 different cancers. Each gene has a binary label indicating whether it is a *transcription factor*.

The experimental setup on the Airport dataset closely resembles that of our synthetic experiments in Table 1, but with real data and more detailed comparisons. We train all models (except for the untrained ones) on the Europe network, and test them on all three networks. The results are presented in Table 2. We notice that the node degree features themselves (with MLP) show reasonable performance in all three networks, which is not surprising since the popularity-based airport role labels are highly relevant to node degrees. The untrained GIN encoder yields a significant margin over both node degrees and the untrained vanilla GCN encoder, indicating the importance of proper aggregation mechanisms. While training of the GCN (through GVAE) and GIN (through DGI) can further improve the performance on the source graph, EGI shows the best performance there with the structure-respecting node degree features (59.15), corroborating the claimed effectiveness of EGI in capturing the essential graph information as we stress in §2.

When transfering the models to USA and Brazil networks, EGI further achieves the best performance compared with all baselines when node degree features are used (64.55 and 73.15), which reflects the most significant positive transfer. Interestingly, direct application of GVAE and DGI without the consideration of essential graph information as we stress leads to rather limited and even negative transferrability (through comparison against the untrained GCN and GIN encoders). The recently

---

[2]We are not exploring graph-level tasks and but focusing on transfer knowledge between two graphs. Thus, we drop the graph-level pre-training tasks in the paper since it is not applicable to our setting.

[3]The downstream tasks are unspecified because we aim to study the general transferability of GNNs that is not bounded to specific tasks. Nevertheless, we assume the tasks to be relevant to graph structures.

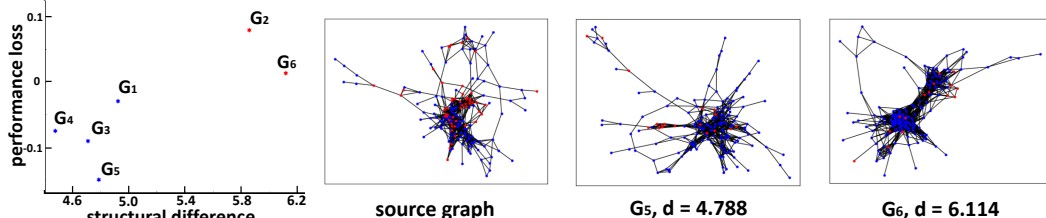

**Figure 2:** Role identification on the Gene dataset. Due to severe label imbalance that vanishes the performance gaps, we only use the 7 brain cancer networks that have a more consistent balance of labels. We visualize the source graph $G_0$ and two example target graphs that are relatively more similar ($G_5$) and different ($G_6$) with $G_0$.

proposed transfer learning frameworks for GNN like Mask-GIN and Structural Pre-train are able to mitigate negative transfer to some extent, but their performances are still inferior to EGI. We believe this is because their models do not aim to capture the underlying ego-graph distributions as we deem important, so they are prune to learn the graph-specific information that is less transferable across different graphs. Similarly as in Table 1, we also compute the structural difference among three networks *w.r.t.* to RHS of Eq. 5. The structural difference is 12.03 between the Europe and USA networks, and 12.14 between the Europe and Brazil datasets, which are pretty close. Consequently, the transferability of EGI regarding its performance gain over the untrained GIN baseline is $4.8\%$ on the USA network and $4.4\%$ on the Brazil network, which are also pretty close. Such observations once again align well with our conclusion in Theorem 2.1 that the transferability of EGI is closely related to the structural different between source and target graphs.

**Table 2:** Results of role identification with direct-transferring on the Airport dataset. The performance reported (%) are the average over 100 runs. The scores marked with [**] passed t-test with p < 0.01 over the second best results. More details about the results and dataset can be found in Appendix §C.2.

| Method | Europe (source) | | USA (target) | | Brazil (target) | |
|---|---|---|---|---|---|---|
| | node degree | uniform | node degree | uniform | node degree | uniform |
| MLP | 52.81 | 20.59 | 55.67 | 20.22 | 67.11 | 19.63 |
| GCN (untrained) | 52.96 | 20.11 | 55.30 | 22.07 | 68.30 | 17.63 |
| GIN (untrained) | 55.75 | 53.88 | 61.56 | 58.32 | 70.04 | 70.37 |
| GVAE (GCN) (Kipf & Welling, 2016) | 53.90 | 21.12 | 55.51 | 22.39 | 66.33 | 17.70 |
| DGI (GIN) (Velickovic et al., 2019) | 57.75 | 22.13 | 54.90 | 21.76 | 67.93 | 18.78 |
| Mask-GIN (Hu et al., 2019a) | 56.37 | 55.53 | 60.82 | 54.64 | 66.71 | 74.54 |
| ContextPred-GIN (Hu et al., 2019a) | 52.69 | 49.95 | 50.38 | 54.75 | 62.11 | 70.66 |
| Structural Pre-train (Hu et al., 2019b) | 56.00 | 53.83 | 62.17 | 57.49 | 68.78 | 72.41 |
| EGI (GIN) | **59.15**[**] | 54.98 | **64.55**[**] | 57.40 | **73.15**[**] | 70.00 |

On the Gene dataset, with more graphs available, we focus on EGI to further analyze the utility of Eq. 5 in Theorem 2.1, regarding the connection between the structural difference of two graphs and the performance gap of EGI on them. As shown in Figure 2, we train EGI on one graph and test it on six different graphs. The $x$-axis shows the structural difference measured *w.r.t.* the RHS of Eq. 5, and $y$-axis shows the performance loss compared with an untrained GIN. The positive correlation between two quantities is obvious. Specifically, when the structural difference is small, positive transfer is observed as the performance of transfered EGI is better than untrained GIN, and when the structural difference becomes large, negative transfer is observed. Note that, at its current stage, Eq. 5 in Theorem 5 mainly gives a relative indication on the transferability of EGI, because the absolute values of structural difference may vary a lot across different datasets.

## 3.2 FEW-SHOT LEARNING ON RELATION PREDICTION

Here we evaluate EGI in the more generalized and practical setting of *few-shot learning* on the less structure-relevant task of relation prediction, with task-specific node features and fine-tuning. The source graph contains a cleaned full dump of 579K entities from YAGO (Suchanek et al., 2007), and we investigate 20-shot relation prediction on a target graph with 24 relation types, which is a sub-graph of 115K entities sampled from the same dump. In *post-fine-tuning*, the models are pre-trained with an unsupervised loss on the source graph and fine-tuned with the task-specific loss on the target graph. In *joint-fine-tuning*, the same pre-trained models are jointly optimized *w.r.t.* the unsupervised pre-training loss and task-specific fine-tuning loss on the target graph. In Table 3, we observe most of the existing models fail to transfer across pre-training and fine-tuning tasks, especially in the *joint-fine-tuning* setting. In particular, both Mask-GIN and ContextPred-GIN rely a

lot on task-specific fine-tuning, while EGI focuses on the capturing of similar ego-graph structures that are transferable across graphs. As a consequence, EGI significantly outperforms all compared methods in both settings.

**Table 3:** Performance of few-shot relation prediction on YAGO. Structural Pre-train (Hu et al., 2019b) can not scale to the YAGO graphs with 100K+ nodes. More details can be found in Appendix §C.3.

| Method | post-fine-tuning | | joint-fine-tuning | |
|---|---|---|---|---|
| | AUROC | MRR | AUROC | MRR |
| No pre-train | 0.6866 | 0.5962 | N.A. | N.A |
| GVAE (Kipf & Welling, 2016) | 0.7009 | 0.6009 | 0.6786 | 0.5676 |
| DGI (Velickovic et al., 2019) | 0.6885 | 0.5861 | 0.6880 | 0.5366 |
| Mask-GIN (Hu et al., 2019a) | 0.7041 | 0.6242 | 0.6720 | 0.5603 |
| ContextPred-GIN (Hu et al., 2019a) | 0.6882 | 0.6589 | 0.5293 | 0.3367 |
| EGI | **0.7389**** | **0.6695** | **0.7870**** | **0.7289**** |

## 4 RELATED WORK

Representation learning on graphs has been studied for decades, with earlier spectral-based methods (Belkin & Niyogi, 2002; Roweis & Saul, 2000; Tenenbaum et al., 2000) theoretically grounded but hardly scaling up to graphs with over a thousand of nodes. With the emergence of neural networks, unsupervised network embedding methods based on the Skip-gram objective (Mikolov et al., 2013) have replenished the field (Tang et al., 2015; Grover & Leskovec, 2016; Perozzi et al., 2014; Ribeiro et al., 2017). Equipped with efficient structural sampling (random walk, neighborhood, *etc.*) and negative sampling schemes, these methods are easily parallelizable and scalable to graphs with thousands to millions of nodes. However, these models are essentially transductive as they compute fully parameterized embeddings only for nodes seen during training, which are impossible to be transfered to unseen graphs.

More recently, researchers introduce the family of graph neural networks (GNNs) that are capable of inductive learning and generalizing to unseen nodes given meaningful node features (Kipf & Welling, 2017; Defferrard et al., 2016; Hamilton et al., 2017). Yet, most existing GNNs require task-specific labels for training in a semi-supervised fashion to achieve satisfactory performance (Kipf & Welling, 2017; Hamilton et al., 2017; Velickovic et al., 2018; Chen et al., 2018), and their usage is limited to single graphs where the downstream task is fixed. To this end, several unsupervised GNNs are presented, such as the auto-encoder-based ones like GVAE (Kipf & Welling, 2016) and GNFs (Liu et al., 2019), as well as the deep-infomax-based ones like DGI (Velickovic et al., 2019) and InfoGraph (Sun et al., 2019). Their potential in the transfer learning of GNN remains unclear when the node features and link structures vary across different graphs.

Although the architectures of GNNs are not very complicated, training a dedicated model for each graph can still be cumbersome (Chen et al., 2018; Ying et al., 2018a). Moreover, as pre-training neural networks are proven to be successful in other domains (Devlin et al., 2019; He et al., 2016), the idea is intriguing to transfer well-trained GNNs from relevant source graphs to improve the modeling of target graphs or enable few-shot learning (Vinyals et al., 2016; Finn et al., 2017; Ravi & Larochelle, 2017) when labeled data are scarce. In the light of this, pioneering works have studied both generative (Hu et al., 2020) and discriminative (Hu et al., 2019a,b) GNN pre-training schemes. Among these work, though Graph Contrastive Coding (Qiu et al., 2020) shares similar structural view as ours, it utilizes contrastive learning in the embedding space instead of structural space as EGI. Unsupervised domain adaptive GCNs (Wu et al., 2020) study the domain adaption problem while source and target tasks are homogenous. Previous pre-training and self-supervised GNNs lack a rigorous analysis towards their transferability and thus have unpredictable effectiveness.

## 5 CONCLUSION

To the best of our knowledge, this is the first research effort towards establishing a theoretically grounded framework to analyze GNN transferability, which we also demonstrate to be practically useful for guiding the design and conduct of transfer learning with GNNs. For future work, it is intriguing to further strengthen the bound with relaxed assumptions, rigorously extend it to the more complicated and less restricted settings regarding node features and downstream tasks, as well as analyze and improve the proposed framework over more transfer learning scenarios and datasets.

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

## A  THEORY DETAILS

From the $\mathcal{L}_{\text{EGI}}$ objective, we have assumed $g_i \overset{i.i.d.}{\sim} \mu$, $x_i \overset{i.i.d.}{\sim} \nu$, and $(g_i, x_i) \overset{i.i.d.}{\sim} p$. Then with graph $G$, we have access to the empirical distributions of the three. So the sampling reduces to bootstrapping in the procedure of evaluating the objective.

Note that, in Eq. 2 of the main paper, we used a $d$ dimensional hidden state $h_{p,q}^{\tilde{q}}$, specified in Eq. 13 to denote an edge encoding derived from the structure of the ego-graph and the associated source node feature from $(p-1)$-th layer. For simplicity, we consider the concatenated vector $f(x^i)\|z_i$, where $f(x^i) = h_{p,q}^{\tilde{q}}\|x_{p,q}^i$ and $h_{p,q}^{\tilde{q}}, x_{p,q}^i$ are as defined in the EGI model and in 13. Additionally, since both of $h_{p,q}^{\tilde{q}}$ and $x_{p,q}^i$ are normalised, $f$ is bounded.

Finally, as we are considering GNN with $k$ layers, its computation only depends on the k-hop ego-graphs of $G$, which is an important consideration when unfolding the embedding of GNN at a centre node.

A.1 PROOF FOR THEOREM 3.1

**Lemma A.1.** *For any $A \in \mathbb{R}^{m \times n}$, where $m \geq n$, and $A$ is a submatrix of $B \in \mathbb{R}^{m' \times n}$, where $m < m'$, we have*

$$\|A\|_2 \leq \|B\|_2.$$

*Proof.* Note that, $AA^T$ is a principle matrix of $BB^T$, *i.e.*, $AA^T$ is obtained by removing the same set of rows and columns from $BB^T$. Then, by Eigenvalue Interlacing Theorem (Hwang (2004)) and the fact that $A^T A$ and $AA^T$ have the same set of non-zero singular values, the matrix operator norm satisfies $\|A\|_2 = \sqrt{\lambda_{\max}(A^T A)} = \sqrt{\lambda_{\max}(AA^T)} \leq \sqrt{\lambda_{\max}(BB^T)} = \|B\|_2$. $\qquad \square$

We restate Theorem 3.1 from the main paper as below.

**Theorem A.2** (GNN transferability). *Let $G_a = \{(g_i, x_i)\}_{i=1}^n$ and $G_b = \{(g_{i'}, x_{i'})\}_{i'=1}^m$ be two graphs. Then denote $L_{g_i}$ as the (normalised) graph Laplacian of $g_i$ $\forall i = 1, \cdots, n$, and let the node features of $g_i$ be structure-respecting and normalized (similarly for $g_{i'}$). Consider GNN $\Psi_\theta$ with $k$ layers and a 1-hop polynomial filter $\phi_\theta$, the empirical performance difference of $\Psi_\theta$ with $\phi_\theta$ evaluated on $\mathcal{L}_{\text{EGI}}$ satisfies*

$$|\mathcal{L}_{\text{EGI}}(G_a) - \mathcal{L}_{\text{EGI}}(G_b)| \leq \mathcal{O}\left(M + \frac{1}{nm} \sum_{i=1}^n \sum_{i'=1}^m \lambda_{\max}(L_{g_i} - L_{g_{i'}})^{1/2}\right), \qquad (6)$$

*where $M$ is a constant dependant on $k$, $\phi_\theta$, $\{L_{g_i}\}$, $\{L_{g_{i'}}\}$, $\{x_i\}$, $\{x_{i'}\}$. In addition, if $\exists U \in O(n \vee m)^4$ s.t.,*

$$U L_{g_i} U^T = Diag(\lambda(L_{g_i})), \quad U L_{g_{i'}} U^T = Diag(\lambda(L_{g_{i'}}))$$

*we have $\mathcal{O}\left(M + \frac{1}{nm} \sum_{i=1}^n \sum_{i'=1}^m \|\lambda(L_{g_i}) - \lambda(L_{g_{i'}})\|_2\right)$, where $\lambda(L_{g_i})$ denotes the ordered eigenvalues of the graph Laplacian of $g_i \in G_a$ (similarly for $g_{i'}$).*

*Proof.* We denote $\sigma_s(t) = \log(1 + e^t)$, the softplus activation function, which is 1-Lipschitz continuous. Now,

$$|\mathcal{L}_{\text{EGI}}(G) - \mathcal{L}_{\text{EGI}}(G')|$$

$$= \left| \frac{1}{n^2} \sum_{i,j=1}^n (\mathcal{D}(g_i, z_j)) - \frac{1}{n} \sum_{i=1}^n (-(-\mathcal{D}(g_i, z_i)) - (\frac{1}{m^2} \sum_{i',j'=1}^m (\mathcal{D}(g_{i'}, z_{j'})) - \frac{1}{m} \sum_{i'=1}^m (-(-\mathcal{D}(g_{i'}, z_{i'})))) \right|$$

$$\leq \frac{1}{n^2 m^2} \sum_{i,j=1}^n \sum_{i',j'=1}^m |\mathcal{D}(g_i, z_j) - \mathcal{D}(g_{i'}, z_{j'})| + \frac{1}{nm} \sum_{i=1}^n \sum_{i'=1}^m |\mathcal{D}(g_i, z_i) - \mathcal{D}(g_{i'}, z_{i'})|$$

$$= \frac{1}{n^2 m^2} \sum_{i,j=1}^n \sum_{i',j'=1}^m A + \frac{1}{nm} \sum_{i=1}^n \sum_{i'=1}^m B.$$

First we consider $B$. Recall that, $V_p(g_i)$ is the set of nodes in layer $p$ of $g_i$,

$$\mathcal{D}(g_i, z_i) = \sum_{p=1}^k \sum_{q=1}^{|V_p(g_i)|} \log(\sigma_{sig}(U^T \tau(W^T[f(x^i)\|z_i]))),$$

where $\sigma_{sig}(t) = \frac{1}{1+e^{-t}}$ is the sigmoid function, $\tau$ is some $\gamma_\tau$-Lipschitz activation function and $[\cdot\|\cdot]$ denotes the concatenation of two vectors. Then we have

$$U^T \tau(W^T[f(x^i)\|z_i]) = U^T \tau(W_1^T f(x^i) + W_2^T z_i).$$

---
[4]$O(n \vee m)$ is the orthogonal group of order $n \vee m$. So we have $L_{g_i}$ and $L_{g_{i'}}$ admitting simultaneous ordered spectral decomposition.

WLOG, assume $d_p = |V_p(g_i)| = |V_p(g_{i'})| \ \forall u = 1, \cdots, k$. In addition, since $\log(\sigma_{sig}(t)) = -\log(1 + e^{-t}) = -\sigma_s(-t)$, which is 1-Lipschitz, it gives

$$B \leq \sum_{p=1}^{k} \sum_{q=1}^{d_p} |U^T \tau \left( W_1^T f(x^i) + W_2^T z_i \right) - U^T \tau \left( W_1^T f(x^{i'}) + W_2^T z_{i'} \right)|$$

$$\leq \gamma_\tau s_U \sum_{p=1}^{k} \sum_{q=1}^{d_p} (\|W_1^T f(x^i) - W_1^T f(x^{i'})\|_2 + \|W_2^T z_i - W_2^T z_{i'}\|_2) \tag{7}$$

$$\leq \gamma_\tau s_U s_W \sum_{p=1}^{k} \sum_{q=1}^{d_p} (\|f(x^i) - f(x^{i'})\|_2 + \|z_i - z_{i'}\|_2),$$

where $s_U$ is the largest singular value of $U$, and similarly $s_W = s_{W_1} \vee s_{W_2}$. Since we assumed the node features are normalised, then $\|f(x^i) - f(x^{i'})\|_2 \leq c_D$.

From Eq. 7, we only care about $x_i$'s embedding obtained from a $k$-layer GNN with 1-hop polynomial (linear in $L$) filter. Inspired by the characterization of GNN from a node-wise view in Verma & Zhang (2019), we similarly denote the embedding of node $x_i \ \forall i = 1, \cdots, n$ in the final layer of the GNN as

$$z_i^k = z_i = \Psi_\theta(x_i) = \sigma(\sum_{j \in \mathcal{N}(x_i)} e_{.j} z_j^{k-1}) \in \mathbb{R}^d,$$

where $e_{.j} = [\phi_\theta(L)]_{.j} \in \mathbb{R}$. We may denote $z_i^\ell \in \mathbb{R}^d$ similarly for $\ell = 1, \cdots, k-1$, and $z_i^0 = x_i \in \mathbb{S}^{d-1}$ the node feature of node $x_i$. With the assumption of GNN stated in the statement, it is clear that only the k-hop ego-graph $g_i$ centered at $x_i$ is needed to compute $z_i^k$ for any $i = 1, \cdots, n$ instead of the whole of $G$. With such observation in mind, let us denote the matrix of node embeddings of $g_i$ at the $\ell$th layer as $(z_{p,q}^{i(\ell)}) \in \mathbb{R}^{|V(g_i)| \times d}$, for $\ell = 1, \cdots, k$; and let $(z_{p,q}^{i(0)}) \equiv (x_{p,q}^i) \in (\mathbb{S}^{d-1})^{|V(g_i)|}$ denote the matrix of node features in the k-hop ego-graph $g_i$. In addition, we denote $(z_{p,q}^{i(\ell)})_{p \leq t}$ to be the submatrix that is obtained by selecting rows that corresponds to $v \in V_p(g_i)$ for $p = 0, \cdots, t \leq k$. Similarly for $g_{i'}$.

Moreover, let us denote $\phi_\theta(L_{g_i}) \equiv [\phi_\theta(L)]_{g_i}$, *i.e.*, the filtered full graph Laplacian of $G$ subsetted by the k-hop ego-graph $g_i$. Then, let $\phi_\theta(L_{g_i})_{p \leq t}$ denotes the submatrix that is obtained by selecting rows and columns that corresponds to $v \in V_p(g_i)$ for $p = 0, \cdots, t \leq k$. Similarly for $g_{i'}$.

Therefore, by Lemma A.1, for any $\ell = 1, \cdots, k$, the following holds

$$\|(z_{p,q}^{i'(\ell)})_{p \leq t} - (z_{p,q}^{i'(\ell)})_{p \leq t}\|_2 \leq \|(z_{p,q}^{i'(\ell)})_{p \leq t+1} - (z_{p,q}^{i'(\ell)})_{p \leq t+1}\|_2.$$

Assume $\|(z_{p,q}^{i'(\ell-1)})\|_2 \leq c_z < \infty \ \forall \ell$. Now, at the final layer,

$$\|z_i - z_{i'}\|_2 = \|(z_{p,q}^{i'(k)})_{p=0} - (z_{p,q}^{i'(k)})_{p=0}\|_2$$

$$\leq \|[\sigma(\phi_\theta(L_{g_i})_{p \leq 1}(z_{p,q}^{i(k-1)})_{p \leq 1}) - \sigma(\phi_\theta(L_{g_{i'}})_{p \leq 1}(z_{p,q}^{i'(k-1)})_{p \leq 1})]_{p=0}\|_2$$

$$\leq \gamma_\sigma \|\phi_\theta(L_{g_i})_{p \leq 1}(z_{p,q}^{i(k-1)})_{p \leq 1} - \phi_\theta(L_{g_{i'}})_{p \leq 1}(z_{p,q}^{i'(k-1)})_{p \leq 1}\|_2$$

$$\leq \gamma_\sigma \|\phi_\theta(L_{g_i})_{p \leq 1}\|_2 \|(z_{p,q}^{i(k-1)})_{p \leq 1} - (z_{p,q}^{i'(k-1)})_{p \leq 1}\|_2 \tag{8}$$

$$+ \gamma_\sigma \|(z_{p,q}^{i'(k-1)})_{p \leq 1}\|_2 \|\phi_\theta(L_{g_i})_{p \leq 1} - \phi_\theta(L_{g_{i'}})_{p \leq 1}\|_2$$

$$\leq \gamma_\sigma \|\phi_\theta(L_{g_i})\|_2 \|(z_{p,q}^{i(k-1)})_{p \leq 1} - (z_{p,q}^{i'(k-1)})_{p \leq 1}\|_2 + \gamma_\sigma c_z \|\phi_\theta(L_{g_i}) - \phi_\theta(L_{g_{i'}})\|_2.$$

In general, for $\ell = 1, \cdots, k-1$, the following holds with $t = k - \ell$,

$$\|(z_{p,q}^{i'(\ell)})_{p \leq t} - (z_{p,q}^{i'(\ell)})_{p \leq t}\|_2$$

$$\leq \gamma_\sigma \|\phi_\theta(L_{g_i})_{p \leq t+1}(z_{p,q}^{i(\ell-1)})_{p \leq t+1} - \phi_\theta(L_{g_{i'}})_{p \leq t+1}(z_{p,q}^{i'(\ell-1)})_{p \leq t+1}\|_2 \tag{9}$$

$$\leq \gamma_\sigma \|\phi_\theta(L_{g_i})\|_2 \|(z_{p,q}^{i(\ell-1)})_{p \leq t+1} - (z_{p,q}^{i'(\ell-1)})_{p \leq t+1}\|_2 + \gamma_\sigma c_z \|\phi_\theta(L_{g_i}) - \phi_\theta(L_{g_{i'}})\|_2.$$

Then we equivalently write Eq. 9 as $E_\ell \leq bE_{\ell-1} + a$, which gives

$$E_\ell \leq b^\ell E_1 + \frac{b^\ell + 1}{b - 1} a.$$

Then, with $(x_{p,q}^i) = (z_{p,q}^{i(0)})$, we see the following is only dependant on the structure of $g_i$ and $g_{i'}$,

$$\|(z_{p,q}^{i'(\ell)}) - (z_{p,q}^{i'(\ell)})\|_2 \le \gamma_\sigma^\ell \|\phi_\theta(L_{g_i})\|_2^\ell \|(x_{p,q}^i) - (x_{p,q}^{i'})\|_2$$
$$+ \frac{\gamma_\sigma^\ell \|\phi_\theta(L_{g_i})\|_2^\ell + 1}{\gamma_\sigma \|\phi_\theta(L_{g_i})\|_2 - 1} \gamma_\sigma c_z \|\phi_\theta(L_{g_i}) - \phi_\theta(L_{g_{i'}})\|_2.$$

Since the features are normalised, and so are the graph Laplacians, we have $\|\phi_\theta(L_{g_i})\|_2 \le c_L$ and $\|(x_{p,q}^i) - (x_{p,q}^{i'})\|_2 \le c_x$. Then with Eq. 8, we have

$$\|z_i - z_{i'}\|_2 \le \gamma_\sigma^k c_L^k c_x + \frac{\gamma_\sigma^k c_L^k + 1}{\gamma_\sigma c_L - 1} \gamma_\sigma \gamma_\theta c_z \|L_{g_i} - L_{g_{i'}}\|_2$$
$$\le c_{\gamma,\Psi}(M + \|L_{g_i} - L_{g_{i'}}\|_2) \tag{10}$$
$$= c_{\gamma,\Psi}(M + \lambda_{\max}(L_{g_i} - L_{g_{i'}})^{1/2}).$$

Now, by Eq. 7, we have

$$B \le k d_{\max} \gamma_\tau s(c_D + c_{\gamma,\Psi}(M + \lambda_{\max}(L_{g_i} - L_{g_{i'}})^{1/2})),$$

where $d_{\max} = \max_p d_p$. Similarly, the above holds for $A$, since from Eq. 8, the node features and embedded features are bounded by separate terms. We therefore arrive at

$$|\mathcal{L}_{\text{EGI}}(G) - \mathcal{L}_{\text{EGI}}(G')| \le 2k d_{\max} \gamma_\tau c_{\gamma,\Psi} s(M' + \frac{1}{nm} \sum_{i=1}^n \sum_{i'=1}^m \lambda_{\max}(L_{g_i} - L_{g_{i'}})^{1/2}))$$
$$\le 2k d_{\max} \gamma_\tau c_{\gamma,\Psi} s(M' + \frac{1}{nm} \sum_{i=1}^n \sum_{i'=1}^m \|L_{g_i} - L_{g_{i'}}\|_F)). \tag{11}$$

Moreover, by Von Neumann's Trace Inequality Grigorieff (1991), if $\exists U \in O(\beta)^5$, where $\beta = \sum_{p=0}^k d_p$, s.t.

$$U L_{g_i} U^T = \text{Diag}(\lambda(L_{g_i})), \quad U L_{g_{i'}} U^T = \text{Diag}(\lambda(L_{g_{i'}})),$$

we have $\|L_{g_i} - L_{g_{i'}}\|_F = \|\lambda(L_{g_i}) - \lambda(L_{g_{i'}})\|_2$, then

$$Eq. \ 11 \le c_{\gamma,\Psi}(M + \|\lambda(L_{g_i}) - \lambda(L_{g_{i'}})\|_2).$$

Therefore Eq. 11 becomes

$$|\mathcal{L}_{\text{EGI}}(G) - \mathcal{L}_{\text{EGI}}(G')| \le 2k d_{\max} \gamma_\tau c_{\gamma,\Psi} s(M' + \frac{1}{nm} \sum_{i=1}^n \sum_{i'=1}^m \|\lambda(L_{g_i}) - \lambda(L_{g_{i'}})\|_2).$$

$\square$

Note that, our view of structural information is closely related to graph kernels (Bai & Hancock, 2016) and graph perturbation (Verma & Zhang, 2019). Specifically, our Def 2.1 is motivated by the concept of k-layer expansion sub-graph in (Bai & Hancock, 2016). However, (Bai & Hancock, 2016) used the Jensen-Shannon divergence between pairwise representations of sub-graphs to define a depth-based sub-graph kernel, while we depict $G$ as samples of its ego-graphs. In this sense, our view is related to the setup in (Verma & Zhang, 2019), which derived a uniform algorithmic stability bound of a 1-layer GNN under 1-hop structure perturbation of $G$.

In the setting of domain adaptation, (Ben-David et al., 2007) draws a connection between the difference in the distributions of source and target domains and the model transferability, and learns a transferable model by minimizing such distribution differences. This coincides with our approach of connecting the structure difference of two graphs in terms of k-hop subgraph distributions and the transferability of GNNs in the above theory.

## B MODEL DETAILS

Following the same notations used in the paper, EGI consists of a GNN encoder $\Psi$ and a GNN discriminator $\mathcal{D}$. In general, the GNN encoder $\Psi$ and decoder $\mathcal{D}$ can be any existing GNN models. For each ego-graph and its node features $\{g_i, x_i\}$, the GNN encoder returns node embedding $z_i$

---

[5]$O(\beta)$ is the orthogonal group of square matrix $\beta$. So we have $L_{g_i}$ and $L_{g_{i'}}$ admits simultaneous ordered spectral decomposition.

for the center node $v_i$. As mentioned in Eq. 2 in the main paper, the GNN discriminator $\mathcal{D}$ makes edge-level predictions as follows,

$$\mathcal{D}(e_{\tilde{v}v}|h_{p,q}^{\tilde{q}}, x_{p,q}^i, z_i) = \sigma\left(U^T \cdot \tau\left(W^T[h_{p,q}^{\tilde{q}}||x_{p,q}^i||z_i]\right)\right), \qquad (12)$$

where $e_{\tilde{v}v} \in E(g_i)$ and $h_{p,q}^{\tilde{q}} \in \mathbb{R}^d$ is the representation for edge $e_{\tilde{v}v}$ between node $v_{p-1,\tilde{q}}$ in hop $p-1$ and $v_{p,q}$ in hop $p$. Specifically, we denote the source node at $p-1$ hop as $\tilde{q} \in \tilde{Q}_{p,q}$, $\tilde{Q}_{p,q} = \{\tilde{q} : v_{p-1,\tilde{q}} \in V_{p-1}(g_i), e_{(p-1,\tilde{q})(p,q)} \in E(g_i)\}$. Hence, the edge prediction relies on the combination of center node embedding $z_i$, destination node feature $x_{p,q}^i$ and edge message $h_{p,q}^{\tilde{q}}$.

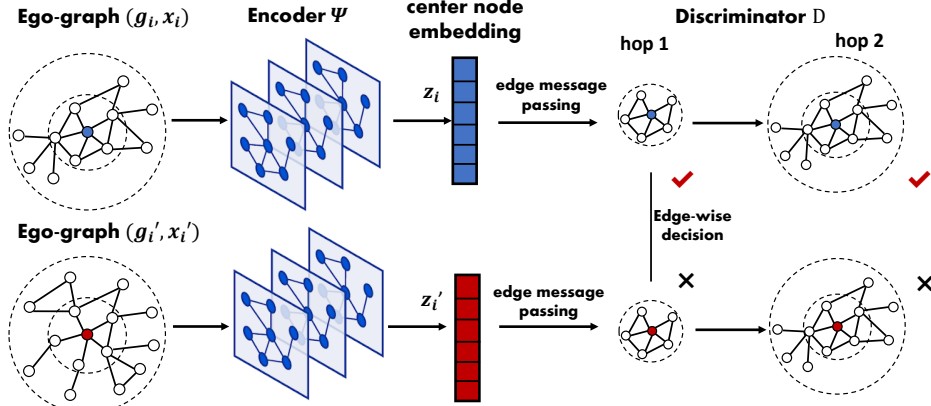

**Figure 3:** The overall EGI training framework.

In Figure 3, $\{g_i, x_i\}$ and $\{g_i', x_i'\}$ are the positive and negative training samples **w.r.t** ego-graph topology $g_i$. The discriminator $\mathcal{D}$ operates on a reversed ego-graph $\tilde{g}_i$ comparing encoder's forward propagation on $g_i$. It starts from the center node $v_i$ and compute the hidden representation $m_{p-1,\tilde{q}}$ for node $v_{p-1,q}$ at each hop. The edge message $h_{p,q}^{\tilde{q}}$ is calculated between source node's hidden representation $m_{p-1,\tilde{q}}$ and destination node features $x_{p,q}$ .

$$h_{p,q}^{\tilde{q}} = \text{ReLU}\left(W_p^T\left(m_{p-1,\tilde{q}} + x_{p,q}^i\right)\right), \; m_{p-1,\tilde{q}} = \frac{1}{|\tilde{Q}_{p-1,\tilde{q}}|} \sum_{q' \in \tilde{Q}_{p-1\tilde{q}}} h_{p-1,\tilde{q}}^{q'} \qquad (13)$$

When $p = 1$, every edge origins from the center node $v_i$ and $m_{0,q'}$ is the center node feature $x_{v_i}$.

In every batch, we sample a set of ego-graphs and their node features $\{g_i, x_i\}$. During the forward pass of encoder $\Psi$, it aggregates from neighbor nodes to the center node $v_i$. Then, the discriminator calculates the edge embedding in Eq. 12 from center node $v_i$ to its neighbors and make edge-level predictions– *fake* or *true*. The training framework of EGI is depicted in Figure 3 and Algorithm 1.

We implement our method and all of the baselines using the same encoders $\Psi$: 2-layer GIN (Xu et al., 2019) for synthetic and role identification experiments, 2-layer GraphSAGE (Hamilton et al., 2017) for the relation prediction experiments. We set hidden dimension as 32 for both synthetic and role identification experiments, For relation prediction fine-tuning task, we set hidden dimension as 256. We train EGI in a mini-batch fashion since all the information for encoder and discriminators are within the k-hop ego-graph $g_i$ and its features $x_i$. Further, we conduct neighborhood sampling and set maximum neighbors as 10 to speed up the parrallel training. The space and time complexity of EGI is $O(BN^K)$, where $B$ is the batch size, $N$ is the number of the neighbors and k is the number of hops of ego-graphs. Notice that both the encoder $\Psi$ and discriminator $\mathcal{D}$ propagate message on the k-hop ego-graphs, so the extra computation cost of $\mathcal{D}$ compared with a common GNN module is a constant multiplier over the original one. The scalability of EGI on million scale YAGO network is reported in section C.3.

### B.1 TRANSFER LEARNING SETTINGS

The goal of transfer learning is to train a model on a dataset or task, and use it on another. In our graph learning setting, we focus on training the model on one graph and using it on another. In particular, we focus our study on the setting of *direct-transfering*, where the model learned on the source graph is directly applied on the target graph without *fine-tuning*. We study this setting because

---

**Algorithm 1:** Pseudo code for training EGI

---

1   The GNN encoder $\Psi$ and the GNN discriminator $\mathcal{D}$, k-hop ego graph and features $\{g_i, x_i\}$;

2   /* EGI-training starts */

3   **while** $\mathcal{L}_{\text{EGI}}$ *not converges* **do**

4      Sample M ego-graphs $\{(g_1, x_1), ..., (g_M, x_M)\}$ from empirical distribution $\mathbb{P}$ without replacement, and obtained their positive and negative node embeddings $z_i, z_i'$ through $\Psi$
$$z_i = \Psi(g_i, x_i), z_i' = \Psi(g_i', x_i'),$$
     /* Initialize positive and negative expectation in Eq. 1 in the main paper*/

5      $E_{pos} = 0, E_{neg} = 0$

6      **for** $p$ = 1 to $k$ **do**

7         /* Compute JSD on edges at each hop*/

8         **for** $e_{(p-1,\tilde{q})(p,q)} \in E(g_i)$ **do**

9            generate edge embedding $h_{p,q}^{\tilde{q}}$ in Eq. (13) ;

10            $E_{\text{pos}} = E_{\text{pos}} + \sigma\left(U^T \cdot \tau\left(W^T[h_{p,q}^{\tilde{q}}||x_{p,q}^i||z_i]\right)\right)$

11            $E_{\text{neg}} = E_{\text{neg}} + \sigma\left(U^T \cdot \tau\left(W^T[h_{p,q}^{\tilde{q}}||x_{p,q}^i||z_i']\right)\right)$

12         **end**

13      **end**

14      /* Compute batch loss*/

15      $\mathcal{L}_{\text{EGI}} = E_{\text{neg}} - E_{\text{pos}}$

16      /* Update $\Psi, \mathcal{D}$ */

17      $\theta_\Psi \xleftarrow{+} -\nabla_\Psi \mathcal{L}_{\text{EGI}}, \theta_\mathcal{D} \xleftarrow{+} -\nabla_\mathcal{D} \mathcal{L}_{\text{EGI}}$

18   **end**

---

it allows us to directly measure the transferability of GNNs, which is not affected by the fine-tuning process on the target graph. In other words, the fine-tuning process introduces significant uncertainty to the analysis, because there is no guarantee on how much the fine-tuned GNN is different from the pre-trained one. Depending on specific tasks and labels distributions on the two graphs, the fine-tuned GNN might be quite similar to the pre-trained one, or it can be significantly different. It is then very hard to analyze how much the pre-trained GNN itself is able to help. Another reason is about efficiency. The fine-tuning of GNNs requires the same environment set-up and computation resource as training GNNs from scratch, although it may take less training time eventually if pre-training is effective. It is intriguing if this whole process can be eliminated when we guarantee the performance with direct-transfering.

In our experiments, we also study the setting of transfer learning with fine-tuning, particularly on the real-world large-scale YAGO graphs. Since we aim to study the general transferability of GNNs not bounded to specific tasks, we always pre-train GNNs with the unsupervised pre-training objective on source graphs. Then we enable two types of fine-tuning. The first one is *post-fine-tuning* ($\mathcal{L} = \mathcal{L}_s$), where the pre-trained GNNs are fine-tuned with the supervised task specific objective $\mathcal{L}_s$ on the target graphs. The second on is *joint-fine-tuning* ($\mathcal{L} = \mathcal{L}_s + \mathcal{L}_u$), where pre-training is the same, but fine-tuning is done *w.r.t.* both the pre-training objective $\mathcal{L}_u$ and task specific objective $\mathcal{L}_s$ on target graphs in a semi-supervised learning fashion. The unsupervised pre-training objective $\mathcal{L}_u$ of EGI is Algorithm 1, while those of the compared algorithms are as defined in their papers. The supervised fine-tuning objective $\mathcal{L}_s$ is the same as in the DistMult paper (Yang et al., 2014) for all algorithms.

## C    EXPERIMENT DETAILS

### C.1    SYNTHETIC EXPERIMENTS

**Data.** As mentioned in the main paper, we use two traditional graph generation models for synthetic data generation: (1) barabasi-albert graph (Barabási & Albert, 1999) and (2) forest-fire graph (Leskovec et al., 2005). We generate 40 graphs each with 100 nodes with each model. We control the parameters of two models to generate two graphs with different ego-graph distributions. Specifically, we set the number of attached edges as 2 for barabasi-albert model and set $p_{\text{forward}} = 0.4$, $p_{\text{backward}} = 0.3$ for forest-fire model. In Figure 4a and 4b, we show example graphs from two families in our datasets. They have the same size but different appearance which leads to our study on the

transferability gap in Table 1 in the main paper. The accuracy of this task defined as the percentage of nearest neighbors for target node in the embedding space that are structure-equivalent, *i.e.* #correct k-nn neighbors / #ground truth equivalent nodes.

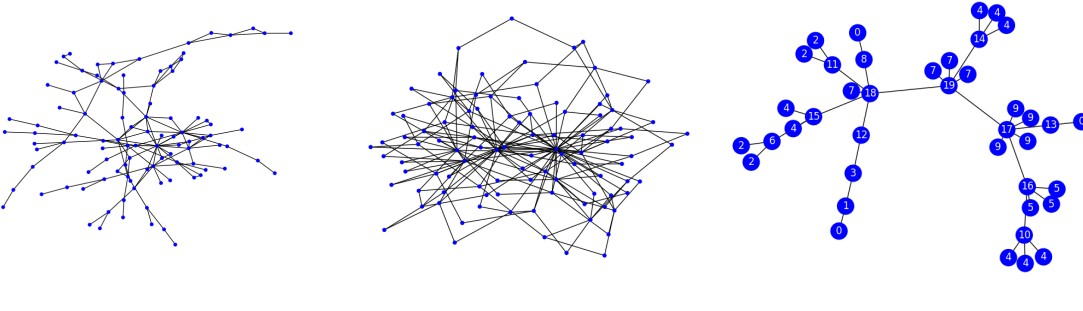

(a) Forest-fire graph example     (b) Barabasi-albert graph example     (c) structural label example

**Figure 4:** Visualizations of the graphs and labels we use in the synthetic experiments.

**Results.** The structural equivalence label is obtained by a 2-hop WL-test (Weisfeiler & Lehman, 1968) on the ego-graphs. If two nodes have the same 2-hop ego-graphs, they will be assigned the same label. In the example of Figure 4c, the nodes labeled with same number (*e.g.* 2, 4) have the isomorphic 2-hop ego-graphs. Note that this task is exactly solvable when node features and GNN architectures are powerful enough like GIN (Xu et al., 2019). In order to show the performance difference among different methods, we set the length of one-hot node degree encoding to 3 (all nodes with degrees higher than 3 have the same encoding). Here, we present the performance comparison with different length of degree encodings (d) in Table 4. When the capacity of initial node features is high (d=10), the transfer learning gap diminishes between different methods and different graphs because the structural equivalence problem can be exactly solved by neighborhood aggregations. However, when the information in initial node features is limited, the advantage of EGI in learning and transfering the graph structural information is obvious. In Table 5, we also show the performance of different transferable and non-transferable features, *i.e.* node embedding (Perozzi et al., 2014) and random feature vectors. The observation is similar with Table 1 in the main paper: the transferable feature can reflect the performance gap between similar and dissimilar graphs while non-transferable features can not.

In both Table 4 and 7 here as well as Table 1 in the main paper, we report the structural difference among graphs in the two sets ($\bar{d}$) calculated *w.r.t.* the term $\frac{1}{nm}\sum_{i=1}^{n}\sum_{i'=1}^{m}\|\lambda(L_{g_i}) - \lambda(L_{g_{i'}})\|_2$ on the RHS of Theorem 2.1 in the main paper. This indicates that the Forest fire graphs are structurally similar to the other Forest fire graphs, while less similar to the Barabasi graphs, as can be verified from Figure 4a and 4b. Our bound in Theorem 3.1 then tells us that the GNNs (in particular, EGI) should be more transferable in the F-F case than B-F. This is verified in Table 4 and 5 when using the transferable node features of degree encoding with limited dimension (d=3) as well as DeepWalk embedding, as EGI trained on Forest fire graphs performs significantly better on Forest fire graphs than on Barabasi graphs (with +0.094 and +0.057 differences, respectively).

**Table 4:** Synthetic experiments of identifying structural-equivalent nodes with different degree encoding dimensions.

| Method | #dim degree encoding d = 3 | | | # dim degree encoding d = 10 | | | structural difference | |
|---|---|---|---|---|---|---|---|---|
| | F-F | B-F | $\Delta$ | F-F | B-F | $\Delta$ | $\bar{d}$(F,F) | $\bar{d}$(B,F) |
| GCN (untrained) | 0.478 | 0.478 | / | 0.940 | 0.940 | / | | |
| GIN (untrained) | 0.572 | 0.572 | / | 0.940 | 0.940 | / | | |
| GVAE (GCN) | 0.498 | 0.432 | +0.066 | 0.939 | 0.937 | 0.002 | 1.78 | 2.17 |
| DGI (GIN) | 0.578 | 0.591 | -0.013 | 0.939 | 0.941 | -0.002 | | |
| EGI (GIN) | **0.710** | 0.616 | +0.094 | 0.942 | 0.942 | 0 | | |

## C.2   REAL-WORLD ROLE IDENTIFICATION EXPERIMENTS

**Data.** We report the number of nodes, edges and classes for both airport and gene dataset. The numbers for the Gene dataset are the aggregations of the total 52 gene networks in the dataset. For

**Table 5:** Synthetic experiments of identifying structural-equivalent nodes with different transferable and non-transferable features.

| Method | DeepWalk embedding | | | random vectors | | | structural difference | |
|---|---|---|---|---|---|---|---|---|
| | F-F | B-F | $\Delta$ | F-F | B-F | $\Delta$ | $\bar{d}$(F,F) | $\bar{d}$(B,F) |
| GCN (untrained) | 0.658 | 0.658 | / | 0.246 | 0.246 | / | | |
| GIN (untrained) | 0.663 | 0.663 | / | 0.520 | 0.520 | / | | |
| GVAE (GCN) | 0.713 | 0.659 | +0.054 | 0.266 | 0.264 | 0.002 | 1.78 | 2.17 |
| DGI (GIN) | 0.640 | 0.613 | +0.027 | 0.512 | 0.576 | -0.064 | | |
| EGI (GIN) | **0.772** | 0.715 | +0.057 | 0.507 | 0.485 | +0.022 | | |

**Table 6:** Overall Dataset Statistics

| Dataset | # Nodes | # Edges | # Classes |
|---|---|---|---|
| Europe | 399 | 5,995 | 4 |
| USA | 1,190 | 13,599 | 4 |
| Brazil | 131 | 1,074 | 4 |
| Gene | 9,228 | 57,029 | 2 |

the three airport networks, Figure 5 shows the power-law degree distribution on log-log scale. The class labels are between 0 to 3 reflecting the level of the airport activities (Ribeiro et al., 2017). For the Gene dataset, we matched the gene names in the TCGA dataset (Yang et al., 2019) to the list of transcription factors on wikipedia[6]. 75% of the genes are marked as 1 (transcription factors) and some gene graphs have extremely imbalanced class distributions. So we conduct experiments on the relatively balanced gene graphs of brain cancers (Figure 2 in the main paper). Both datasets do not have organic node attributes. The role-based node labels are highly relevant to their local graph structures, but are not trivially computable such as from node degrees.

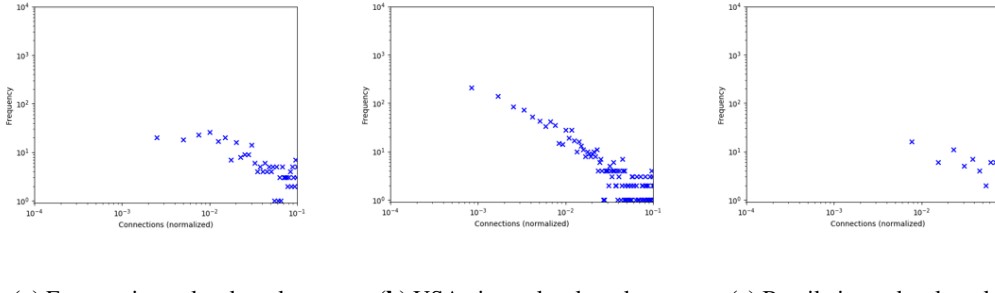

**(a)** Europe airport log-log plot   **(b)** USA airport log-log plot   **(c)** Brazil airport log-log plot

**Figure 5:** Visualizations of power-law degree distribution on three airport dataset.

**Results.** As we can observe from Figure 5, the three airport graphs have quite different sizes and structures (*e.g.*, regarding edge density and connectivity pattern). Thus, the absolute classification accuracy in both Table 2 in the main paper and Table 7 here varies across different graphs. However, as we mention in the main paper, the structural difference we compute based on Eq. 5 in Theorem 3.1 is close among the Europe-USA and Europe-Brazil graph pairs (12.03 and 12.14), which leads to close transferability of EGI from Europe to USA and Brazil. This indicates the effectiveness of our view over essential structural information.

Note that, the results present in Table 7 are the accuracy of GNNs directly trained and evaluated on each network without transfering. Therefore, only the Europe column has the same results as in Table 2 in the main paper, while the USA and Brazil columns can be regarded as providing an upper-bound performance of GNN transfered from other graphs. As we can see, EGI gives the closest results from Table 2 in the main paper to Table 7 here, demonstrating the its plausible transferability. The scores are so close, showing a possibility to skip fine-tuning when the source and target graphs are similar enough. Also note that, although the variances are pretty large (which is also observed

---

[6]https://en.wikipedia.org/wiki/Transcription_factor

in other works like (Ribeiro et al., 2017) since the networks are small), our t-tests have shown the improvements of EGI to be significant.

**Table 7:** Role identification that identifies structurally similar nodes on real-world networks. The performance reported are the average and standard deviation for 10 runs. Our classification accuracy on three datasets all passed the t-test (p<0.01) with the second best result in the table.

| Method | Airport (Ribeiro et al., 2017) | | |
|---|---|---|---|
| | Europe | USA | Brazil |
| node degree | 52.81% ± 5.81% | 55.67% ± 3.63% | 67.11% ± 7.58% |
| GCN (random-init) | 52.96% ± 4.51% | 56.18% ± 3.82% | 55.93% ± 1.38% |
| GIN (random-init) | 55.75% ± 5.84% | 62.77% ± 2.35% | 69.26% ± 9.08% |
| GVAE (GIN) (Kipf & Welling, 2016) | 53.90% ± 4.65% | 58.99% ± 2.44% | 55.56% ± 6.83% |
| DGI (GIN) (Velickovic et al., 2019) | 57.75% ± 4.47% | 62.44% ± 4.46% | 68.15% ± 6.24% |
| Mask-GIN (Hu et al., 2019a) | 56.37% ± 5.07% | 63.78% ± 2.79% | 61.85% ± 10.74% |
| ContextPred-GIN (Hu et al., 2019a) | 52.69% ± 6.12% | 56.22% ± 4.05% | 58.52% ± 10.18% |
| Structural Pre-train (Hu et al., 2019b) | 56.00% ± 4.58% | 62.29% ± 3.51% | 71.48% ± 9.38 % |
| EGI (GIN) | **59.15% ± 4.44%** | **65.88% ± 3.65%** | **74.07% ± 5.49%** |

## C.3 REAL-WORLD LARGE-SCALE RELATION PREDICTION EXPERIMENTS

**Data.** As shown in Table 8, the source graph we use to pre-train GNNs is the full graph cleaned from the YAGO dump (Suchanek et al., 2007), where we assume the relations among entities are unknown. The target graph we use is a subgraph uniformed sampled from the same YAGO dump (we sample the nodes and then include all edges among the sampled nodes). The similar ratio between number of nodes and edges can be observed in Table 8. On the target graph, we also have the access to 24 different relations (Shi et al., 2018) such as *isAdvisedBy*, *isMarriedTo* and so on. Such relation labels are still relevant to the graph structures, but the relevance is lower compared with the structural role labels. We use the 256-dim degree encoding as node features for pre-training on the source graph, then we use the 128-dim positional embedding generated by LINE (Tang et al., 2015) for fine-tuning on the target graph, to explicitly make the features differ across source and target graphs.

**Results.** In Section B.1, we introduced two different types of fine-tuning, *i.e.*, *post-fine-tuning* and *joint-fine-tuning*. For both types of fine-tuning, we add one feature encoder $\mathcal{E}$ before feeding it into the GNNs for two purposes. First, the target graph fine-tuning feature usually has different dimensions with the pre-training features, such as the node degree encoding we use. Second, the semantics and distributions of fine-tuning features can be different from pre-training features. The feature encoder aims to bridge the gap between feature difference in practice. The supervised loss used in this experiment is the same as in DistMult (Yang et al., 2014). In particular, the bilinear score function is calculated as $s(h, r, t) = z_h^T M_r z_t$, where $M_r$ is a diagonal matrix for each relation $r$, $z_h$ and $z_t$ the the embedding of GNN encoder $\Psi$ for head and tail entities. The experiments were run on GTX1080 with 12G memories. We report the average training time per epoch of our algorithm in pre-training and fine-tuning stage in Table 8 as well. The pre-training and fine-tuning takes about 40 epochs and 10 epochs to converge, respectively. In Table 8, we also present the per-epoch training time of EGI. EGI takes about 338 seconds per epoch for optimizing the ego-graph information maximization objective on YAGO-source. As we can see, fine-tuning also takes significant time compared to pre-training, which strengthens our arguments about avoiding or reducing fine-tuning through structural analysis. We implement all baselines within the same pipeline, and the runtimes are all at the same scale.

**Table 8:** dataset statistics and running time of EGI

| Dataset | # Nodes | # Edges | # Relations | # Train/Test | Training time per epoch |
|---|---|---|---|---|---|
| YAGO-Source | 579,721 | 2,191,464 | / | / | 338 seconds |
| YAGO-Target | 115,186 | 409,952 | 24 | 480/409,472 | 134 seconds |

