# OpenReview forum: "Transfer Learning of Graph Neural Networks with Ego-graph Information Maximization"
_ICLR.cc/2021/Conference — Reject_

### Official Review · AnonReviewer1 · 2020-10-25
**Ego-graph information maximization (EGI) is more transferable GNN**

**Rating:** 4
**Confidence:** 5

**Review:**

The authors propose ego-graph information maximization (EGI) to build more transferable GNN. They further theoretically study the transferability of EGI.


The article is fairly well structured, apart from the literature review, which is somewhat missing significant related works, such as Ron Levie's papers, which are the first theoretical studies regarding the transferability of GCNs. Concerning terminology, a reminder of what is called “node features” would be very useful for the reader, as this paper seems to construct features based on the graph structure instead of node attributes or signal over nodes as the node features. The article has the unfortunate tendency to contains a few unproven claims. For instance, "we establish a theoretically grounded and practically useful framework for the transfer learning of GNNs", which would deserve some empirical/theoretical evidence. Yet, the paper is only exploring the transferability of the proposed GIN.

The main argument is to establish a theoretically grounded framework for the transfer learning of GNNs and leverage it to propose a practically transferable GNN model, which I think is a very valid motivation for this paper and also is an important problem, but I think the authors could not achieve this goal properly. From my point of view, this paper proposes a new GNN model and further show theoretically and empirically that their new model is more transferable in some specific circumstances though, it’s quite hard to see some conclusions in my opinion based on both real and simulated experiments (but the transferability arguments of EGI holds to some degree nonetheless).

Comments/concerns/questions:

- In practice, the dataset consists of signals defined on many different graphs; the trained GNN should generalize to signals on graphs unseen in the training set. However, the experiments did not consider such a situation and only focused on some specific constructed features. Also, the authors claim the node features should be a function of the structure. The node attributes (signals over graph) can somehow be considered a noisy function of the graph structure. I would like to see the comparison performance in such situations too, where the graph has node attributes by itself.

- What kind of functions over the graph is reasonable to construct the node features? How much noise is acceptable?

- While the authors talk about the joint distribution of graph structure and node features, I think they are only using graph structure in two ways. Then, the main question is how one can use the graph structure as node attributes. Can one concatenate different constructed node features as node attributes? For example, assuming using both node degree and one-hub embedded graph based on VGAE.

- It has been claimed "[...] under an analogous setting of domain adaptation[..]". How can one interpret it as domain adaptation? Can you show the embedded space?

- Considering the graph convolutional neural network, either "On the Transferability of Spectral Graph Filters" or  "Transferability of Spectral Graph Convolutional Neural Networks" can be considered one important baseline.

- Does $\psi$ is the same for all graphs? If not, the heterogeneous embedding spaces should be an issue.

- The similarity is not interpretable, and it is dependent on data. This is one of the main drawbacks of this paper. Specifically, for EGI on the Gene dataset, the structural difference higher than 5.4 causes a negative transfer effect; however, a structural difference larger than 12 can improve the airport dataset performance. In practice, we do not know the true answer. How can one know if the method is practical in a new dataset?

- Besides that, $G_5$ has a higher distance than $G_4$ and $G_3$; however, it exploits the transferability of EGI more. A discussion on this phenomenon is needed.

- The empirical results do not show the same relationship between DGI and the structural differences. If one considers the transferable feature column for DGI, please compare the performance improvement with the untrained GIN.

- I believe the 3rd and 4th row of the second table is wrong. Please double check. Based on the first table, DGI outperforms EGI for non-transferable features. A comparison, based on another kind of features are needed.

- Table 2 needs to have other columns similar to the first table to see how $\Delta$ will be.

- I also would like to see the target graph B performance in Table 1 (F-B and B-B). Also, the performance of considering two other airports as the source for the airport dataset.

- Comparison with the baselines on the Gene dataset would be helpful.

- Comparing EGI with SOTA GNN methods in some other analytical tasks, including node classification, link prediction, etc. can help validate this model's capability as a new GNN.

- Including a simulation based on some specific perturbations on the target graph is also insightful.

------- UPDATE ---------

I thank the authors for their response to my concerns/comments. It seems like I have to defend my position for suggesting a rejection of the paper. While the response of the authors has clarified some aspects, some comments have not been adequately addressed.

R1: However, the authors could not show how to measure the structural differences between the source and target graphs? Their measurement is not stable and is different between different datasets. How one can evaluate a new dataset is good for transferring or not (R5)? You could see how Ron Levie et al. showed this in their experiments.

R2: But embedding using GCN is another definition for the function of the structure. However, it does not work well as node degree. Therefore, the author cannot point out that you only need to construct the features based on a function of the structure.

R4: You can somehow use their idea to show how much difference is reasonable to transfer information.

R5: So it seems this is one of the main drawbacks of considering \delta. If that is the case, you need to discuss it in the paper.

R7: I believe the response is not sufficient. The authors need to add that to show they could positively transfer information.

---

> ### Author Response · Authors · 2020-11-15
> **Response to Reviewer1 (1/2)**
>
> We first thank the reviewer for the valuable comments and we believe it will help us improve the technical quality of the submission. We summarize the major questions and provide detailed clarifications below.
>
> Q1: What is the goal of this paper (the paper is only exploring the transferability of the proposed GNN; The article is fairly well structured, apart from the literature review, which is somewhat missing significant related works, such as Ron Levie's papers, which are the first theoretical studies regarding the transferability of GCNs).
>
> The goal of this paper is to first establish the requirements of a transferable GNN, then develop EGI as such a GNN, and finally study its transferability wrt the structural difference between source and target graphs. Therefore, yes we are only rigorously studying the transferability of the proposed EGI model, while we clearly show that existing GNNs do not have such transferability both in concept and through experiments.
>
> In the seminal paper by Ron Levie et al, the transferability is described as the robustness of the filter to small graph perturbations and re-indexing of the vertices. It is rather different from what we aim to study, that is, the performance difference of a GNN on a pair of source-target graphs.
> Moreover, it analyzed the transferability of GCN with the goal of calibrating the filter learnt on the discretized domain (i.e. graphs), and compared it to that on the underlying continuum domain (i.e. manifold). On the other hand, we assume a space of k-hop ego-graphs, and analyze the direct transfer performance with our proposed objective, which is shown to be dependent on the difference of ordered local spectrum of two graphs (domains). We will add such discussions in an updated draft.
>
>
>
> Q2: What is the specific requirement for the transferable node features (a.k.a structure-respecting features) ? Can we use specific node attributes? How much noise is acceptable?
>
> From Definition 2.3, we suggest to use structure-respecting node features to study GNN transferability, such as node degree or spectral embedding features, which in an ideal case should correspond to an injective function. The effectiveness of these features has been shown in Table 1, 4 and 5 in our original draft. As for node features other than those directly computed from graph structures (either not structure-respecting or rather noisy), it is simply too challenging to study them all together with the various graph structures in one paper. In this work, we design EGI and focus on the rigorous study on its transferability wrt graph structures as a first step. In the future, it would be interesting to study the specific measures for structure-respecting (thus transferable) node features and acceptable noises, and rigorously analyze them through controlled experiments regarding random and adversarial disturbance.
>
> Q3: Why is our approach analogous to the domain adaptation problem?
>
> When we mentioned domain adaptation, both our setting and bound are similar with the work  (Ben-David et al.).  In that paper, the transferability is measured as a generalization bound on target when applying a classifier trained in the source domain (without fine-tuning). We refer to this strategy as “direct-transfering” in the paper since many pre-training GNN papers actually heavily rely on fine-tuning in the target domain with labels. Besides, the GNN transferability in our Theorem 2.1 can be seen as a difference in empirical risk between source and target ego-graph samples (similarly, the domain adaptation paper studies expected risk on target). It is different from the literature of domain invariant representation learning referred to by the reviewer.
>
> Q4: Consider the GCN transferability of Ron Levie et al. as a baseline in the experiments.
>
> In the paper ‘Transferability of Spectral Graph Convolutional Neural Networks’ by Ron Levie et al, the authors propose a theoretical setting of their transferability objective. In particular, they have studied the robustness of Cayleynet to noise on one real world network, under the supervised learning setting. As stated in the response to Q1, we mainly study the performance difference on two different real graphs, under the setting of unsupervised learning. Without prior knowledge on the prediction tasks on the target graph, we compare different self-training objectives rather than different filters. Hence, the comparison with Cayleynet is possible but not our focus due to such different settings. We have, on the other hand, conducted experiments towards empirical learning problems and compared the performances on multiple down-stream tasks.

---

> ### Author Response · Authors · 2020-11-15
> **Continued Response to Reviewer1 (2/2)**
>
> Q5: Table 2 needs to have other columns similar to the first table to see how $\Delta$ will be.
> $\Delta$  is not interpretable and it is dependent on the data.
>
> The $\Delta$ terms for Table 2 are put in the textual paragraph above Table 2 (12.03, 12.14) due to the space limit.
>
> We would like to note that the upper bound term in theorem 2.1 is an indicator for structurally similar/different source/target graphs, which is not a distance metric. It is an absolute empirical loss but it only enables relative comparisons of transferability in the one-source-multiple-targets or multiple-sources-one-target settings. For example, if we have one target and multiple possible sources at hand, the measure allows us to select the best source to train the model, which transfers best to the target. The upper bound is calculated between pairs of local graph Laplacians from source and target graphs. The variation of $\Delta$ across different   domains is due to the different number of nodes in ego graphs. The airport graphs have more dense hub nodes, which lead to a higher $\Delta$ value than those on the gene graphs.
>
> To obtain “normalized” $\Delta$ values across different domains/datasets, one shall assume the same inference model as well as the same number of maximal neighbors in the k-hop graphs. Under this setting, we re-calculate the bound term for airport dataset as (5.11, 5.10) between Europe-USA and Europe-Brazil, respectively, which are closer to those indicating the positive transfers in the gene dataset. In an updated draft, we would like to clarify that the $\Delta$ term can not be compared across different datasets unless the inference model and parameters are the same.
>
> Q6: Experimental results are counter intuitive between GVAE and GIN(untrained). Based on the first table, DGI outperforms EGI for non-transferable features. A comparison, based on other kinds of features, is needed.
>
> The 3rd and 4th row in Table 2 in the original draft are correct.  Graph Isomorphism Network (GIN) [1] yields superior performance on structure relevant tasks, such as graph classification. Therefore, an untrained GIN can outperform an untrained GCN and even a trained GVAE in a transfer learning setting, since the role identification task as described in section 3.1 is highly structure relevant and GVAE itself is ineffective in the transfer learning setting.
>
> The main purpose of presenting the performances of all algorithms on the non-transferable features is to show that, in general, they perform much worse compared with themselves on the transferable features, which supports our requirement for GNN transferability on structure-respecting node features (Def 2.2). There is no point in comparing the relative performances among the algorithms on the non-transferable features, since the scores are basically too bad to be informative.
>
> Q7: I also would like to see the target graph B performance in Table 1 (F-B and B-B). Also, the performance of considering two other airports as the source for the airport dataset.
>
> The purpose of comparing F-F and F-B is to verify the motivation and effectiveness of the proposed transferability bound (better transferability between more structurally similar graphs). We have listed synthetic experiments with different features and more results on the airport datasets in the Appendix. We will also add the experiments as suggested by the reviewer in an updated draft.
>
> Q8: Comparing EGI with SOTA GNN methods in some other analytical tasks.
>
> Aside from the node classification, we also provide a link prediction experiment in the paper (YAGO), and EGI outperforms other pre-training GNNs significantly. Although there are more different designs in SOTA GNNs, these designs are mostly task driven. In this work, we focus on the transferability of general GNN encoders and thus only compare with the widely used ones like GCN and GIN. In the future, it would be promising to apply the EGI objectives on task-specific SOTA GNNs as we demonstrate the effectiveness of joint-finetuning, where we can add an EGI loss atop of task-oriented GNN losses.
>
> Q9: Does $\Psi$  is the same for all graphs? If not, the heterogeneous embedding spaces should be an issue.
>
> In this paper, we study the homogenous graphs in the experiments. The proposed EGI is a transferable training objective rather than a novel graph filter (encoder) $\Psi$. For heterogeneous graphs, we can combine the EGI objective with the state-of-the-art heterogeneous graph encoders $\Psi$ from the literature.
>
> [1] Xu, Keyulu, et al. "How Powerful are Graph Neural Networks?." International Conference on Learning Representations. 2018.

---

### Official Review · AnonReviewer4 · 2020-10-28
**need some reorganization**

**Rating:** 6
**Confidence:** 3

**Review:**

The paper introduces a theoretical framework for analyzing GNN transferability. The main idea is to view a graph as subgraph samples with the information of both the connections and the features. Based on this view, the authors define EGI score of a graph as a learnable function that needs to be optimized by maximizing the mutual information between the subgraph and the GNN output embedding of the center node.  Then, the authors give an upper bound for the difference of EGI scores of two graphs based on the difference of eigenvalues of the graph Laplacian of the subgraph samples from the two graphs. The implication is that if the difference of the eigenvalues is small, then the EGI scores are similar, which means the GNN has a similar ability to encode the structure of the two graphs.

The idea is new to me. One suggestion is that there are multiple forward references without precise pointers in the paper. Section 2.1 frequently refers to contents in Section 2.2. The authors may want to reorganize the paper to avoid any confusion.

---

> ### Author Response · Authors · 2020-11-15
> **Response to Reviewer4**
>
> We thank the reviewer for the positive feedback. In an updated draft, we will try to reorganize Section 2 by moving forward Table 1 to reduce forward references. For contents that are hard to reorganize (for example, our Def 2.1 and 2.2 have to be introduced before Theorem 2.1), we will at least make the references more precise and easy to follow.

---

### Official Review · AnonReviewer3 · 2020-10-30
**interesting results but a limited coverage**

**Rating:** 6
**Confidence:** 4

**Review:**

The authors proposed a transfer learning scheme for graph neural networks. The proposed method ego-graph information maximization allows learning transferable models. The authors studied structure-respecting node features and provided a theoretical analysis of the transferability of GNNs. The proposed method significantly outperforms state-of-the-art methods.

Clarity:
Overall, this paper reads fine. There are some typos and missing definitions of symbols, e.g., `sp' in eq 1 and $U^T$ in eq 3. D function is defined by another D in eq 2. In definition 2.3, 'Ordered' ego-graph is not defined. 'Title 2.2. ANALYSI', 'structural equivalence', and 'structural different' are typos. The average structural difference denoted by $\bar{d}(,)$
are not defined. The clarity of this manuscript needs to be improved.

Strengths/Quality/Significance (pros):
The interesting observation that the functions learned by GNNs can be viewed as functions to map a subgraph centered at a node to a class label since most GNNs have a few layers and their receptive field of a node output is a k-hop ego-graph.

The authors studied structure-respecting node features, e.g., degrees, spectral embeddings, to show that graph filters of GNNs is transferable. Based on the structure-respecting node features, the authors provide the analysis of transferability solely depending on the graph structure. The analysis showed that the performance gap of transferred models is bounded by a function of the ordered eigenvalues of the graph Laplacian of ego-graphs.

The proposed method achieved significant improvement against baseline approaches.

Weaknesses (cons) & Questions:

The writing should be improved. The manuscript should be self-contained. As mentioned above, there are functions, and variables that are introduced without definitions such as reconstruction loss, sp, $\bar{d}(,)$ and so on.

The analysis is limited to graph structures. To benefit most GNNs in real-world applications, the transferability of GNNs needs to be analyzed with node features as well.

In this paper, the analysis of the transferability of GNNs is limited to node classification. It is not clear whether the proposed method is effective in other tasks on graphs such as link prediction, graph classification.

Even in the synthetic experiments, the performance gain is obtained only in the transferrable feature settings.

--- Post Rebuttal ---
I read the author response and I keep the original rating due to the limited operating range of the proposed method.

---

> ### Author Response · Authors · 2020-11-15
> **Response to Reviewer3**
>
> We first thank the reviewer for the valuable comments and we believe it will help us improve the technical quality of the submission.
>
> Q1: There are functions, and variables that are introduced without definitions such as reconstruction loss, sp, d¯(,) and so on.
>
> Thanks for the careful check and we would like to provide some clarifications. The reconstruction loss described in Eq. 4 can have different forms. For example, in GVAE, $R(g_i | z) = \sigma(z z^T)$, where $z$ is the encoder output. In the paper, different from GVAE, we also mentioned that EGI assumes the edges in an ego-graph to be observed jointly. The specific form of reconstruction in EGI is more complicated as the sample space grows with O(|V|!) and that is the reason for us to optimize the mutual information instead. Similar to DIM [1], sp denotes the softplus function used in the Jensen-Shannon MI estimator. In our controlled synthetic experiments, the results reported are averaged over 40 target graphs. $\bar{d}$(,) used in Table 1 is the average structural difference between one source and multiple target graphs. We will clarify these terms and others in an updated draft.
>
> Q2: The analysis is limited to graph structures. To benefit most GNNs in real-world applications, the transferability of GNNs needs to be analyzed with node features as well.
>
> In this work, we formulate the intuitive requirement on node features as structure-respecting and give some examples of such features. However, it is simply too challenging to study various graph structures together with arbitrary node features towards GNN transferability in one paper. In this work, we design EGI and focus on the rigorous study on its transferability mainly wrt graph structures as a first step. In the future, it would be interesting to study the specific measures for general structure-respecting (thus transferable) node features, and rigorously analyze them through more experiments. Currently, even under the simple manually constructed structure-respecting features, we discovered existing pre-trained/self-training GNN methods like DGI and Mask-GNN fail to transfer (Table 1 and 2), where our EGI consistently performs better with significant margins.
>
>
> Q3: In this paper, the analysis of the transferability of GNNs is limited to node classification. It is not clear whether the proposed method is effective in other tasks on graphs such as link prediction, graph classification.
>
> Without any assumption on target tasks, the transferability of EGI in Theorem 2.1 is analyzed under the EGI objective. In the experiments, we examine the GNN utility in both node classification (Section 3.1) and link prediction (Section 3.2). EGI works pretty well under both two scenarios. For graph classification, EGI is optimized upon ego-graphs and not suitable for graph classification without a “readout” function [2].
>
>
> Q4: Even in the synthetic experiments, the performance gain is obtained only in the transferrable feature settings.
>
> Such results are exactly as we expect, which supports our requirements of using structure-respecting features to ensure the transferability of GNNs (Def 2.2).
>
> [1] Hjelm, R. Devon, et al. "Learning deep representations by mutual information estimation and maximization." International Conference on Learning Representations. 2018.
>
> [2] Sun, Fan-Yun, et al. "InfoGraph: Unsupervised and Semi-supervised Graph-Level Representation Learning via Mutual Information Maximization." International Conference on Learning Representations. 2019.

---

### Official Review · AnonReviewer2 · 2020-11-04
**Solid theoretical analysis and extensive experimental studies**

**Rating:** 7
**Confidence:** 3

**Review:**

This work considers the unsupervised learning for graph neural networks. The work has solid theoretical analysis and extensive experimental studies. To encode the structure information, the K-hop ego-graph is used to generate a k-hop ego-graph for each node. I don’t see any major issues in this work. Here are several small concerns:

1.	In this work, the ordered k-hop ego-graph is used but didn’t discuss how this order is generated. Would the authors explain how the ordering works.
2.	Since k is an important hyper-parameter, can authors provide some experiments to evaluate the impact of different k values.

---

> ### Author Response · Authors · 2020-11-15
> **Response to Reviewer2**
>
> We first thank the reviewer for the valuable comments and we believe it will help us improve the technical quality of the submission.
>
> Q1:In this work, the ordered k-hop ego-graph is used but didn’t discuss how this order is generated. Would the authors explain how the ordering works.
>
> In practice, we use the BFS-ordering  (i.e. the order generated by a BFS traversal with ascending node ids) and we specified this above Eq.2 on Page 3 in the original draft. Different orderings do not affect the propagation results in EGI.
>
> Q2: Since k is an important hyper-parameter, can authors provide some experiments to evaluate the impact of different k values.
>
> EGI is composed by a GNN encoder and an ego-graph discriminator. The number of k mostly affects the expressive power of the GNN encoder, which similarly affects all baselines as well, so we didn’t list it as a hyper-parameter to study in the experiments.
> As noticed by existing works [1, 2], the over-smoothing phenomenon is severe with a large number of layers. In our experiments, we observe k=2 to generally yield the best performance for all compared algorithms. k=3 does not bring significant performance improvements but takes much longer time for training, while k=1 performs much worse due to limited expressive power.
>
> [1] Li, Qimai, Zhichao Han, and Xiao-Ming Wu. "Deeper insights into graph convolutional networks for semi-supervised learning." AAAI 2018.
> [2] Oono, Kenta, and Taiji Suzuki. "Graph neural networks exponentially lose expressive power for node classification." ICLR 2019.

---

### Decision · Program_Chairs · 2021-01-07
**Final Decision**

**Decision:**

Reject

**Comment:**

The paper presents a novel framework from transfer learning over GNNs. Experiments ought to better substantiate how structural differences/similarities are measured, as well as relying on prior art to measure transferability success. A plan for incorporating (structural) features would also strengthen the present work.